# COVID-19 vaccination data management and visualization systems for improved decision-making: Lessons learnt from Africa CDC Saving Lives and Livelihoods program

Raji Tajudeen‡, Mosoka Papa Fallah‡, John Ojo, Tamrat Shaweno *,
Michael Sileshi Mekbib‡, Frehiwot Mulugeta, Wondwossen Amanuel⊙, Moses Bamatura,
Dennis Kibiye, Patrick Chanda Kabwe, Senga Sembuche, Ngashi Ngongo,
Nebiyu Dereje, Jean Kaseya

Africa Centers for Disease Control and Prevention, Addis Ababa, Ethiopia

‡ These authors are co-first authors on this work.
* AdewoT@africacdc.org

## Abstract

The DHIS2 system enabled real-time tracking of vaccine distribution and administration to facilitate data-driven decisions. Experts from the Africa Centres for Disease Control and Prevention (Africa CDC) Monitoring and Evaluation (M&E) and Management Information System (MIS) teams, with support from the Health Information Systems Program South Africa (HISP-SA), developed the continental COVID-19 vaccination tracking system. Several variables related to COVID-19 vaccination were considered in developing the system. Three-hundred fifty users can access the system at different levels with specific roles and privileges. Four dashboards with high-level summary visualizations were developed for top leadership for decision-making, while pages with detailed programmatic results are available to other users depending on their level of access. Africa CDC staff at different levels with a role-based account can view and interact with the dashboards and make necessary decisions based on the COVID-19 vaccination data from program implementation areas on the continent. The Africa CDC vaccination program dashboard provided essential information for public health officials to monitor the continental COVID-19 vaccination efforts and guide timely decisions. As the impact of COVID-19 is not yet over, the continental tracking of COVID-19 vaccine uptake and dashboard visualizations are used to provide the context of continental COVID-19 vaccination coverage and multiple other metrics that may impact the continental COVID-19 vaccine uptake. The lessons learned during the development and implementation of a continental COVID-19 vaccination tracking and visualization dashboard may be applied across various other public health events of continental and global concern.

**Data availability statement:** All data are in the manuscript files.

**Funding:** The Saving Lives and Livelihood program was supported by funding from the Mastercard Foundation. The funders did not play any role in study design, data collection, analysis or manuscript preparation/publication.

**Competing interests:** The authors have declared that no competing interests exist.

**Abbreviations:** Africa CDC, Africa Centres for Disease Control and Prevention; AEFIs, Adverse Effects Following Immunizations; BSR, business specification requirements; CSV, Comma Separated Values; HTML, Hypertext Markup Language; IP, Implementing Partner; JSON, JavaScript Object Notation; KPIs, Key Performance Indicators; M&E, Monitoring and Evaluation; NCs, National Coordinators; SLL, Saving Lives and Livelihoods; COVAX, COVID-19 Vaccines Global Access; WHO, World Health Organization.

## Author summary

In our work, we developed a real-time tracking system for COVID-19 vaccination across Africa, using the DHIS2 platform to help guide data-driven decisions. This system, created by experts from the Africa CDC in collaboration with the Health Information Systems Program South Africa (HISP-SA), enables us to monitor and manage COVID-19 vaccine distribution and administration continent-wide. With nearly 300 users from various levels of Africa CDC, the system allows authorized personnel to access relevant vaccination data tailored to their role, making the information easy to interpret and act upon.

Our system includes four interactive dashboards that provide high-level summaries for top leadership and detailed views for program teams. These visual tools empower Africa CDC staff to make informed decisions, track progress, and address challenges in real-time. As COVID-19 remains a global concern, our platform provides crucial insights into vaccine coverage and related health metrics, demonstrating the potential for similar systems to enhance public health responses to future emergencies across Africa and beyond.

## Introduction

The COVID-19 pandemic has highlighted the need for innovative digital health interventions, particularly in vaccines, due to challenges like global inequities, misinformation, and vaccine hesitancy [1]. The rapid evolution of public health emergencies (PHEs) demands efficient, real-time solutions to support data-driven decision-making and improve response efforts [2]. Digital health technologies have significantly enhanced global health security by enabling timely data collection and analysis, identifying infectious disease trends, and reducing infection risk through remote services [3]. Digital tools including data dashboards are being used extensively in the pandemic, collating real-time public health data, including confirmed cases, deaths, and testing figures, to keep the public informed and support policymakers in refining interventions [4]. Digital tools have a critical role in increasing the efficiency and effectiveness of the COVID-19 vaccine delivery process and the management of the vaccine program. The speed at which the vaccine is being delivered and administered requires the support of digital technologies that can play a critical role in facilitating the planning, delivery, monitoring, and management of vaccination programs [5]. Despite the evident advantages, several challenges hinder the development and implementation of digital solutions in Africa during PHEs. Many African countries face challenges related to internet connectivity, power supply, and access to digital devices, which can impede the widespread adoption of digital health interventions. Ensuring the confidentiality and security of health data remains a significant challenge, particularly with the growing threats of cyberattacks and unauthorized data breaches. Moreover, fragmented health information systems and a lack of standardized protocols hinder seamless data integration across different platforms and institutions [6].

The Saving Lives and Livelihoods (SLL) program, a partnership between the Africa CDC and the Mastercard Foundation, has established information system to provide near real-time (NRT) COVID-19 vaccination data for decision-making [7]. At the beginning of the COVID-19 vaccination rollout, Africa CDC encountered significant challenges in coordinating and monitoring efforts across Member States due to the absence of a centralized, real-time data system. Data on vaccine distribution, uptake, and stock levels were often fragmented, delayed, or inconsistently reported, limiting Africa CDC's ability to track progress, identify gaps, and provide timely technical support. The lack of standardized digital tools also made it difficult to consolidate information, hindering strategic decision-making and the efficient allocation of vaccines and resources. These challenges underscored the urgent need for a unified digital platform to enable accurate, timely, and actionable insights for effective pandemic response. The Africa CDC and HISP South Africa developed a DHIS2-based system to aid in the distribution and use of COVID-19 vaccines in African countries. Notably, COVID-19 dashboards have also been effectively used in different settings to track COVID-19 cases, deaths, and testing throughout the COVID-19 pandemic [8–14]. As the world prepares for future emerging and re-emerging pandemics, there is a need to establish strengthened proactive case identification, efficient triaging protocol, and reporting systems using innovative digital technologies in regions where outbreaks are likely to occur, to increase swift response to future epidemics before they evolve into pandemics [15]. Integrating digital health solutions with immunization strategies holds immense potential for improving immunization coverage and monitoring, especially in the post-COVID-19 era [16]. Strengthening Africa's digital health infrastructure can help improve real-time surveillance, enhance cross-border coordination, and ensure equitable access to healthcare interventions. Herein, we describe the overall process and critical lessons learned in the development and implementation of the Africa CDC SLL COVID-19 vaccination dashboard, highlighting its role in improving vaccination monitoring and public health response in Africa while emphasizing the importance of scaling up digital solutions for future public health threats.

## Methods and materials

### Data sources

COVID-19 vaccination data is sourced from multiple sources, including SLL implementing partners (IPs), Member States, and other public sources such as the weekly COVID-19 logistics data from the World Health Organization (WHO). The DHIS2 platform collects and visualizes this data, providing insights into program implementation and vaccine administration [17]. Implementing Partners, contracted under the SLL program, share data validated by the health authority of the implementing Member State. Data entry and submission to Africa CDC are done weekly, monthly, and quarterly, with data reporting on vaccination figures, total number of vaccine doses administered by the SLL program, risk communication and community engagement, and safety surveillance, respectively. Each SLL IP supports programmatic activities in a Member State, and the uploaded data undergoes approval by designated Africa CDC National Coordinators (NCs) (Fig 1).

### Contents and process of the dashboard development

We used an open-source web-based software platform called DHIS2 software [18] for the dashboard development. DHIS2 was selected as the preferred platform due to its open-source nature, flexibility, and widespread use across African countries for routine health data reporting. Its adaptability to diverse programmatic needs, strong global community, and the presence of local technical expertise in many Member States facilitated rapid adoption and localization. By leveraging an already familiar system, Africa CDC ensured alignment with existing national health information infrastructures, promoted interoperability, and enabled efficient scale-up of digital solutions across the continent. The dashboard was designed in phases, beginning with the design of data collection forms and reporting datasets, followed by the creation of data visualization dashboards, and culminating in modules for data sharing, interpretation, and use, with subject matter experts meeting weekly to draft business specification requirements (BSR). The BSR was then handed over to HISP-SA. The initial draft was approved for internal trial use, data quality assurance checks, and large-area server security checks. The final version was ready for deployment within 3 months.

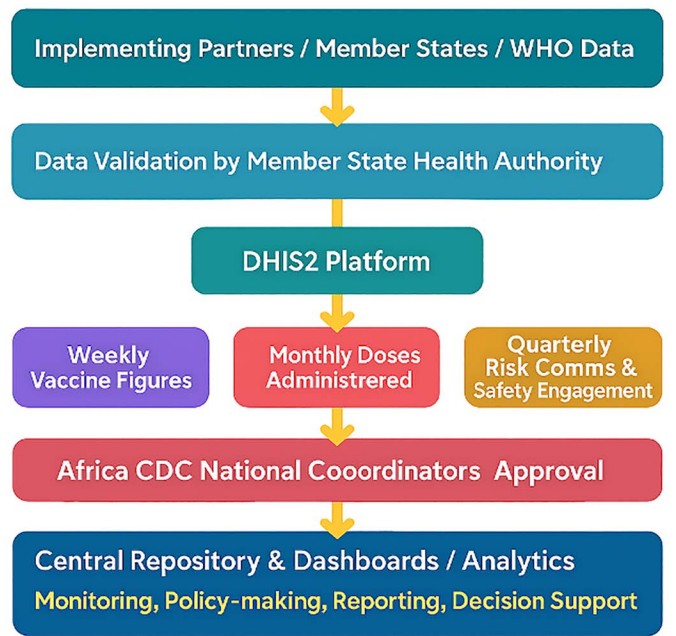

**Fig 1. Schematic diagram of the SLL COVID-19 vaccination reporting system.** The COVID-19 vaccination dashboard presents detailed information on vaccine administration, partnership activities, health worker training, vaccine safety, RCCE, job creation, and project management-related data. The dashboard visually organizes data according to the user and uses standardized Key Performance Indicators (KPIs) to measure performance against targets. Member States with low performance are the focus of the Daily War Room meetings to identify the root causes and seek alternative strategies for improved performance.

The dashboard explores and summarizes vaccination data and attributes, including vaccination coverage, number of persons vaccinated disaggregated by gender, vaccine types used, and risk communication and community engagement (RCCE) activities for Member States implementing the SLL program. The dashboard is structured around 38 datasets, containing 459 distinct data elements and 137 indicators. The dashboards include metrics such as trained vaccinators, vaccine stocks, vaccines nearing expiry, AEFI data, trained community mobilizers, community leaders engaged in delivering COVID-19 prevention messages, etc. (Table 1). The dashboard uses API connectivity for efficient data sharing.

### Access to online data and validation process

The Africa CDC SLL DHIS2 dashboard employs a robust authentication system that grants data access and system functionalities strictly according to user authorization levels. Top leadership members have access to four comprehensive dashboard views designed to support high-level decision-making, while other staff members have access to a limited, role-appropriate subset of these views to maintain data security and focus. Data entry privileges are strictly limited to authorized national-level personnel, with no data entry ability granted to other managerial or staff roles to ensure data integrity and control. Vaccination data is entered through the dedicated "Data Entry App" and undergoes systematic validation using the "Run validation" function before being finalized in the system. This tiered access framework balances data transparency with confidentiality and promotes reliable data management practices across all levels of the program (Fig 2).

### Timelines for uploading COVID-19 vaccination data into the DHIS2 system

The DHIS2 system manages COVID-19 vaccination data, including weekly COVID-19 Vaccination Centres (CVCs) data, monthly RCCE data, and quarterly pharmacovigilance data. CVC data comes from fixed health facilities and mobile

**Table 1. Categories and metrics included in SLL COVID-19 Vaccination Dashboard.**

| Category | Metrics |
|---|---|
| COVID-19 vaccine administration | • Country doses administered weekly<br>• Doses administered monthly<br>• Doses administered by SLL-supported CVCs<br>• Doses administered per population<br>• Doses administered per target population<br>• Partner - Total doses administered by the SLL program<br>• Target of doses administered<br>• Vaccination status by sex<br>• Total number of CVCs<br>• Doses received<br>• Doses wasted<br>• Doses in the stock (AstraZeneca, J&J, Moderna, Pfizer/BioNTech, Sinopharm, Sinovac)<br>• Ancillary units received<br>• Target number of doses to be administered |
| RCCE | • Community health workers performing RCCE<br>• Community health workers performing RCCE at CVC site<br>• People reached through RCCE activities via communication channel<br>• People trained on RCCE approaches and the COVID-19 vaccine<br>• RCCE area of support<br>• Partner community health workers trained<br>• Partner vaccinators trained<br>• Number of COVID-19 vaccination mass campaign conducted<br>• RI sites that have integrated COVID-19 vaccination into PHCs<br>• Partner-Number of vaccination teams supporting campaigns |
| Staffing | • Total number of staffs working onsite<br>• Country data managers trained<br>• Frontline healthcare workers eligible for COVID-19 vaccination<br>• Jobs created by SLL<br>• People trained by National Societies<br>• Youth employment |
| Statistics | • Fully vaccination rate<br>• Proportion of wastage of PEE during transportation<br>• Share of COVID-19 cases sequenced<br>• Share of the population fully vaccinated<br>• Share of target population fully vaccinated<br>• Number of COVID-19-related technical reports published<br>• Scientific presentations in technical meetings |
| AEFI | • AEFIs reported by SLL partners<br>• Confirmed AEFIs following vaccination<br>• Deaths from AEFIs<br>• Reported or suspected (AEFI and adverse AEFI) following COVID-19 vaccination<br>• Serious AEFI reported by SLL partners<br>• Serious AEFIs investigated within 48 hours |
| Procurement | • Total requests received/monthly/quarterly<br>• Vaccine doses delivered<br>• Doses procured by the SLL program<br>• Country number of doses received |

vaccination centres and is reported weekly. National Coordinators review data submissions and either query or approve anomalous data. The system locks in the values to prevent further amendments. NCs analyze partners' data submissions, troubleshoot, and identify anomalous values. RCCE data is submitted on the 15th of the next Month, and NCs review and lock in values. If anomalous data is found, NCs engage IPs to try to understand the root causes and correct

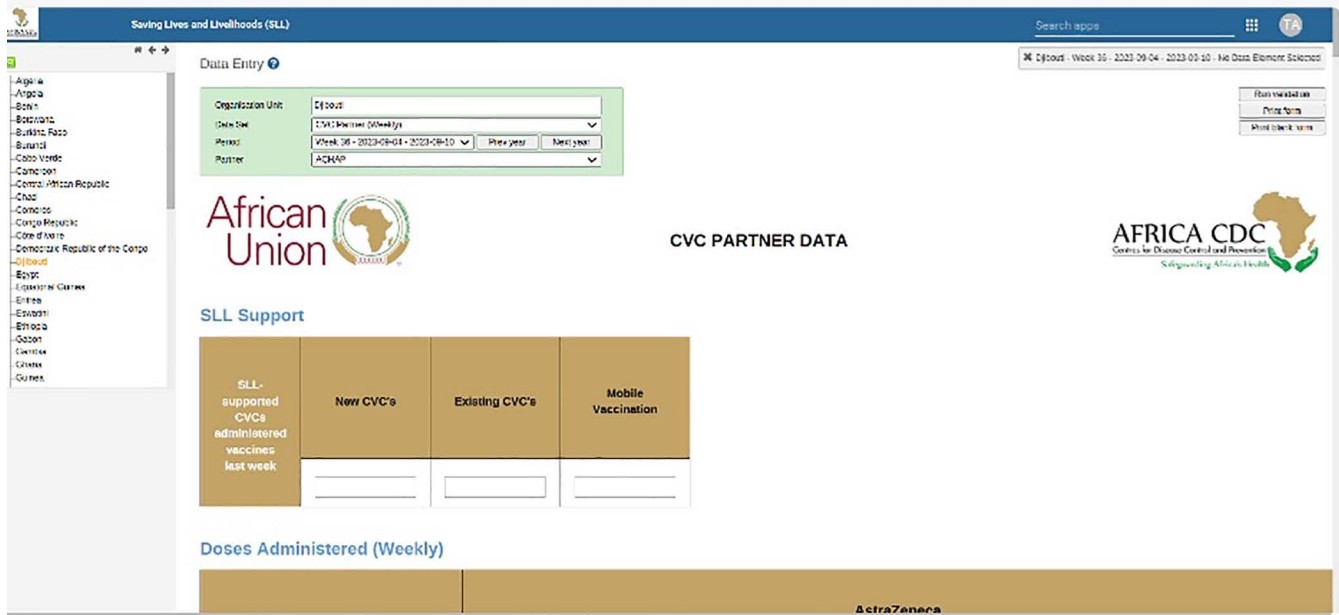

**Fig 2. The online Africa CDC COVID-19 vaccination data entry form.**

the data., DHIS2 automatically verifies data completeness and computes reporting timeliness based on pre-programmed information.

Data is displayed in dashboards within hours, and if data is not shared within the set timeline, an escalation cascade is initiated. If data inconsistencies are detected, a non-compliance state is triggered. Follow-ups are made via email and notifications are sent to NCs in the respective countries to explore the root cause.

## Data visualization, and decision-making process

The Africa CDC and Mastercard Foundation use the DHIS2 system to track and monitor data inputted into the COVID-19 vaccination program. The system features data entry, approval processes, and data visualization apps shown in (Fig 3).

The visualization can be viewed as pivot tables, columns, stacked columns, bars, stacked bars, and line graphs, and can be downloaded as images, tables, Excel data, or other types of formats for further analysis. COVID-19 data visualization improves the digitalization and quality of data and enabling better data use and performance monitoring in Member States. It avoids report delay and disintegrated reporting systems, preventing tedious work of aggregation and human error. The enhanced visualization system has successfully rectified discrepancies observed in data between platforms. Different types of charts available in DHIS2 that can be used to visualize COVID-19 data are shown in (Fig 4).

## Ethical consideration

The project on real-time COVID-19 vaccine tracking via the DHIS2 system did not require ethical approval. The initiative involved creating a digital platform for monitoring vaccination efforts across Africa, with data aggregated for public health decision-making by authorized Africa CDC personnel. As no personally identifiable or sensitive individual-level health data were collected, and users operated within defined professional roles, the project did not qualify as human subject research, hence, an ethical waiver was appropriate.

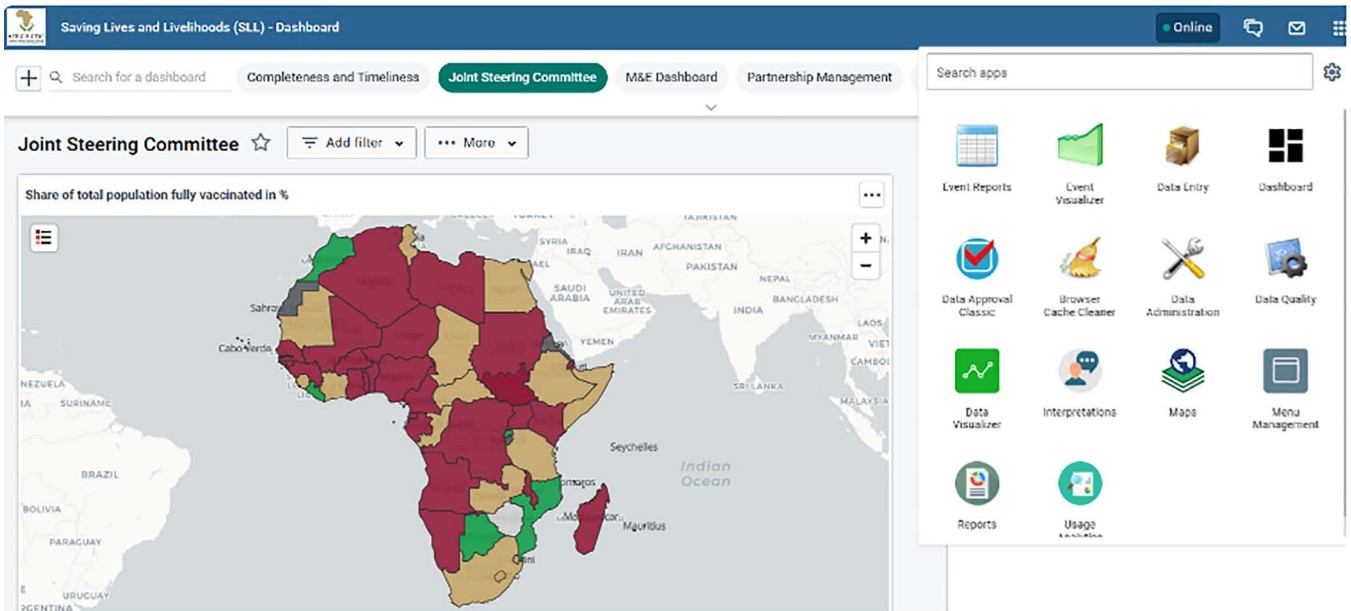

**Fig 3. Key features of Africa CDC SLL Program DHIS2 system Apps: data entry, data approval process, and data visualizer apps.** (Map data Copyright and License: https://www.openstreetmap.org/copyright). The Data Visualizer app allows users to create various charts directly within DHIS2, with specific layout restrictions and use cases.

## Result

### The decision-making process

There are four high-level decision-making bodies in the SLL program, each supported by a dedicated dashboard containing critical information for rapid decision-making. These dashboards, the Joint Steering Committee Dashboard, the Program Management Unit (PMU) Daily War Room Dashboard, the Vaccine Taskforce Dashboard, and the Partnership Management Dashboard enable timely, data-driven actions across program operations. By utilizing the SLL DHIS2 reporting and dashboard system, the four high-level decision-making bodies harmonized and collaborated their decision-making processes, resulting in more coordinated, efficient, and evidence-based actions across Member States.

Since Quarter 4 2021, the dashboards have facilitated substantial public health impact across implementing Member States. Approximately 35 million COVID-19 vaccine doses have been administered, with over 70% coverage of the target population reached in 12 Member States and eight Member States had achieved 100% (Fig 5). The integration of real-time monitoring and data-sharing agreements with 15 Member States allowed decision-makers to urgently reallocate doses, preventing nearly 2 million doses from expiring. Investments in logistics, including the procurement and deployment of 44,000 cold-chain equipment units, were guided by insights from the dashboards, ensuring optimal distribution and utilization. In parallel, 38,000 health workers were trained and deployed, and 23,000 jobs were created, including 25% for youth and 55% for women.

Overall, the SLL DHIS2 dashboards have not only strengthened operational efficiency but also directly informed policy and resource allocation decisions, demonstrating the critical role of digital data systems in large-scale immunization program success.

### Joint Steering Committee Dashboard

The Joint Steering Committee, led by Africa CDC and the Mastercard Foundation, is the highest-level decision-making entity for the SLL program. It oversees program implementation tracking and uses the Joint Steering Committee

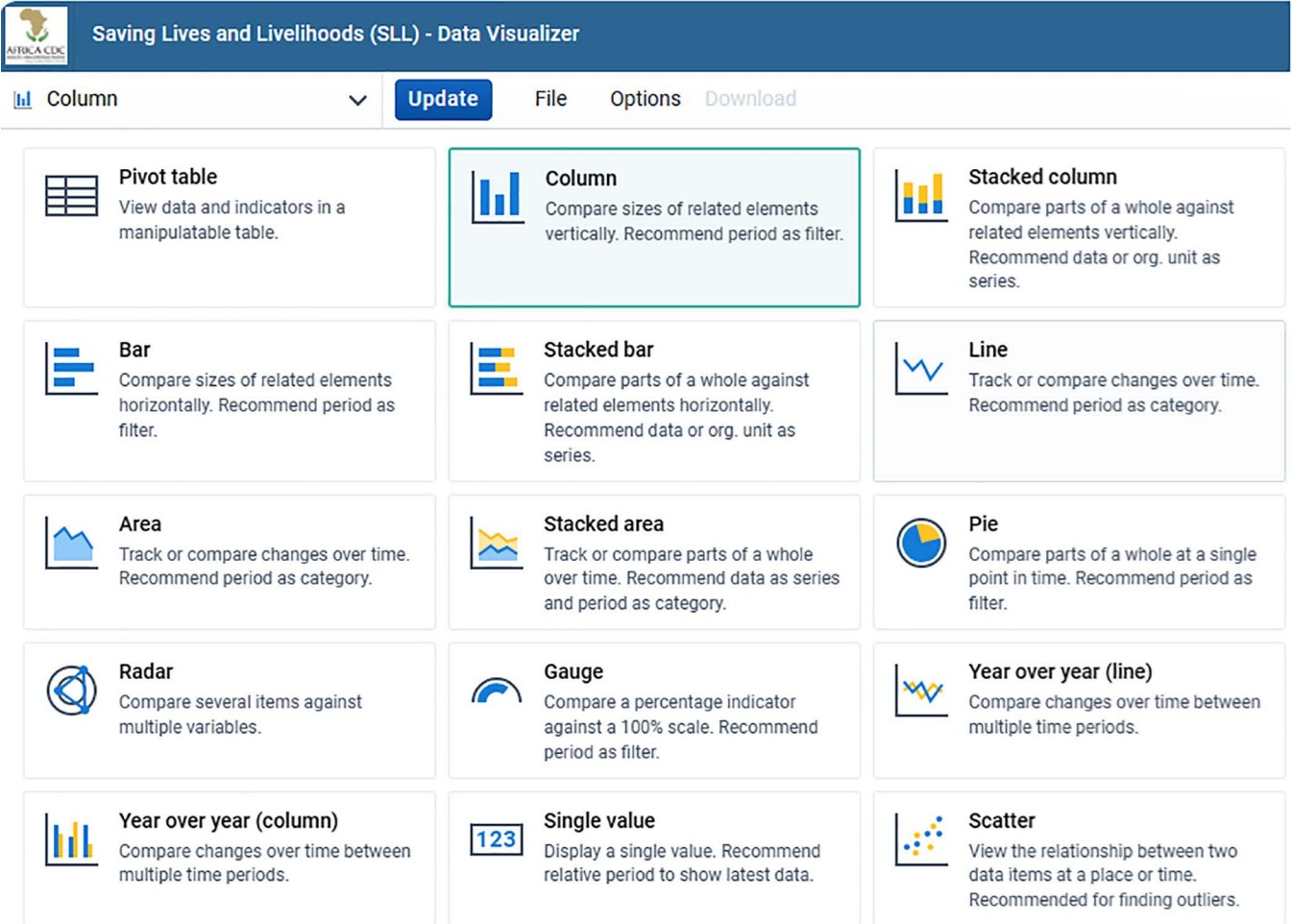

**Fig 4. Types of data visualization apps available in DHIS2.**

dashboard to make timely corrections based on indicators. The dashboard monitors and analyzes overarching SLL and Member State performance against targets such as the share of the total population vaccinated on the continent (S1 Table). An example map shows (Fig 6) that eight Member States had achieved 100% full vaccination of their population, 12 achieved 70%, and the remaining countries had below 70%. The dashboard allows the committee to make actionable decisions across the program sites to ensure Member States with poor performance are compared to better-performing ones.

**Program Management Unit (PMU) Daily War Room**

The PMU is responsible for coordinating the SLL program, monitoring performance, and reporting to the governing bodies. It holds meetings to develop plans, manage relationships with the Mastercard Foundation, support teams, and serve as the single source of truth. The PMU Daily War Room dashboard assesses regional performance tracking performance against targets, and displays weekly target doses to be administered vs Partner doses administered in each implementing Member State. It also shows vaccine administration per existing and mobile CVCs for the last 12 weeks.

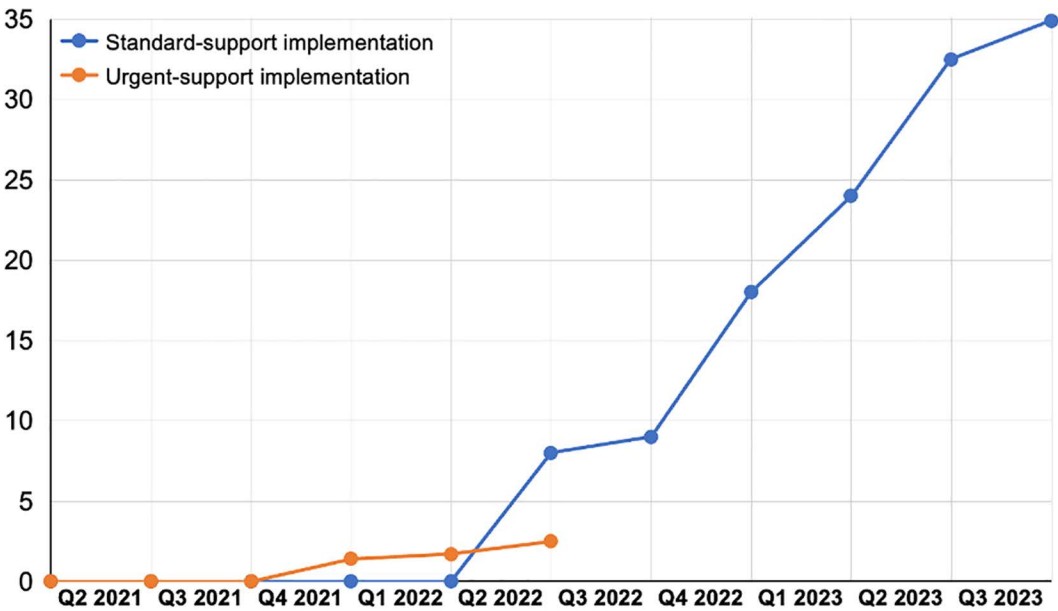

**Fig 5. Cumulative doses administered in Member States.**

### Vaccine Taskforce dashboard

The main objective of the Vaccine Taskforce dashboard is to monitor the overall continental and implementing Member States' performance against targets to enable course correction and make data-driven decisions.

### Partnership Management dashboard

The Partnership Management team is responsible for setting up partners' agreements and supporting partners to clarify contract-related queries during implementation. The team performs activities to include selecting partners and setting up partners' agreements, supporting partners to define their high-level scope of work and interaction model, monitoring compliance-related or contractual issues encountered by partners during implementation, and overseeing sourcing and procurement by partners. The main objective of the Partnership Management dashboard is to monitor partner performance across each Member State (Fig 8).

### Monitoring and evaluation dashboard

Effective performance monitoring and evaluation is done by collaborating with all parties involved and viewing data collection and validation as a joint responsibility. Africa CDC's monitoring and evaluation team oversees and ensures each stakeholder plays a role in collecting, reporting, and validating data and activity pictures. Moreover, the M&E dashboard tracks system performance and user feedback for regular maintenances and troubleshooting. Users, including data managers and National Coordinators, highlighted the dashboard's value in providing real-time visibility that simplified decision-making during vaccine supply fluctuations and campaign surges. However, they noted challenges such as the need for additional training in data visualization and a desire for more granular reporting filters to support local health district planning. Routine War Room meetings facilitated timely troubleshooting of data anomalies and offered contextualized support when dashboard analytics identified persistent reporting gaps or vaccine wastage issues. This feedback has been instrumental in continuously improving system usability and data quality for enhanced program management.

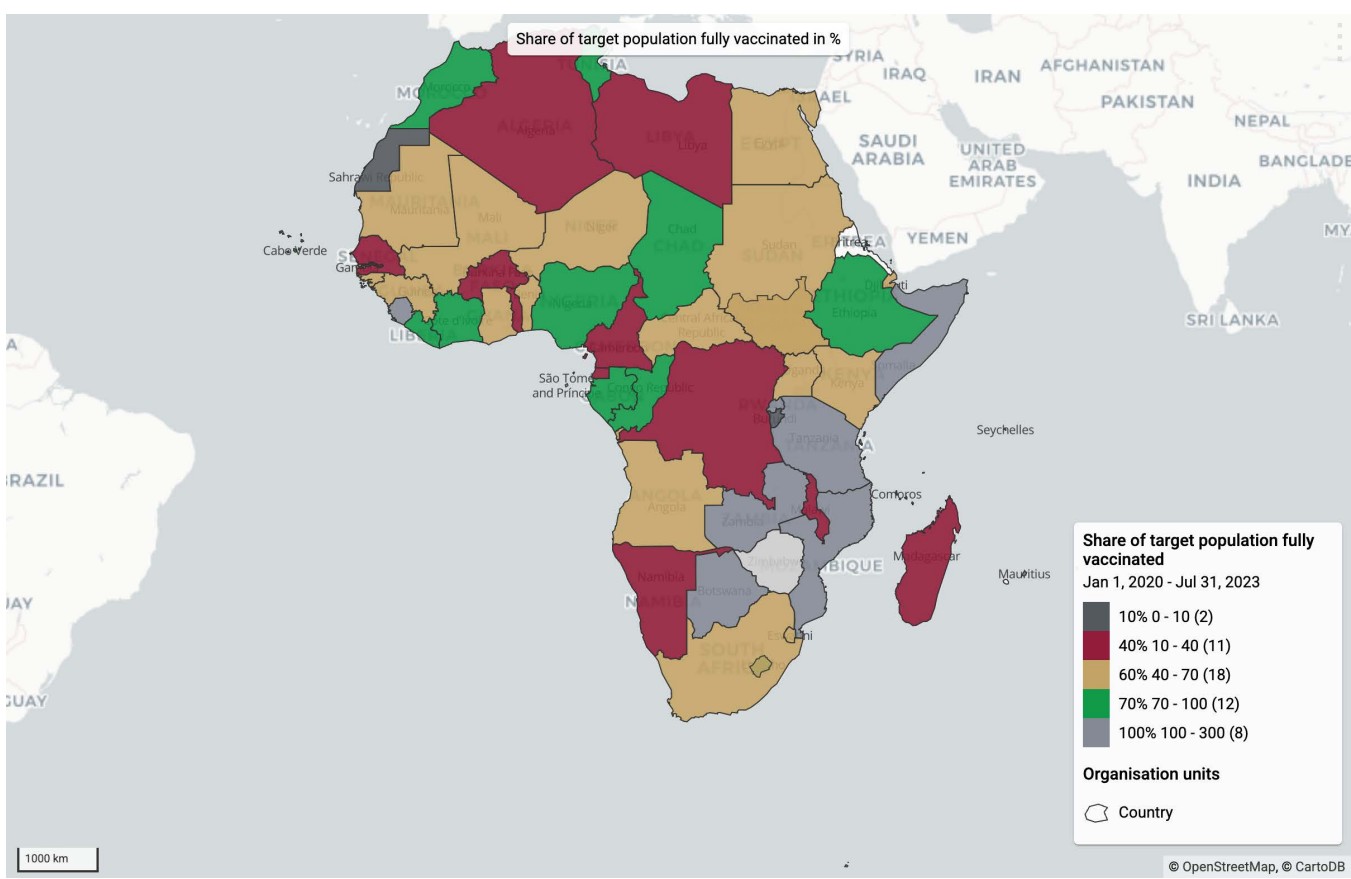

**Fig 6. Share of target population fully vaccinated in percentage (Map data Copyright and License: https://www.openstreetmap.org/copyright).** The Joint Steering Committee can use the vaccination dashboards disaggregated by region. The below outputs show the cumulative performance against targets in one of the five African Union regional coordinating centers, the Eastern AU region (S2 Table). From the dashboard in (Fig 7), the overall regional performance against the target as well as individual country performance against the target can be easily tracked, and timely decisions can be made.

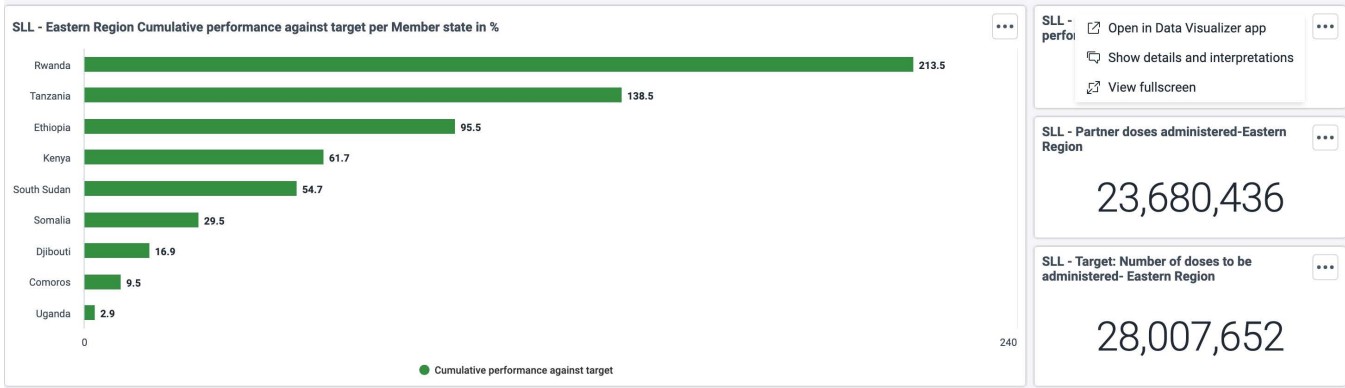

**Fig 7. SLL - Eastern Region Cumulative performance against target per Member state in %.** The Joint Steering Committee dashboard displays a chart indicating the Saving Lives and Livelihoods program's contribution to COVID-19 vaccination administration in each Member State. From the chart, SLL contributed 31.2% of vaccines administered in Rwanda, followed by Ethiopia (26%).

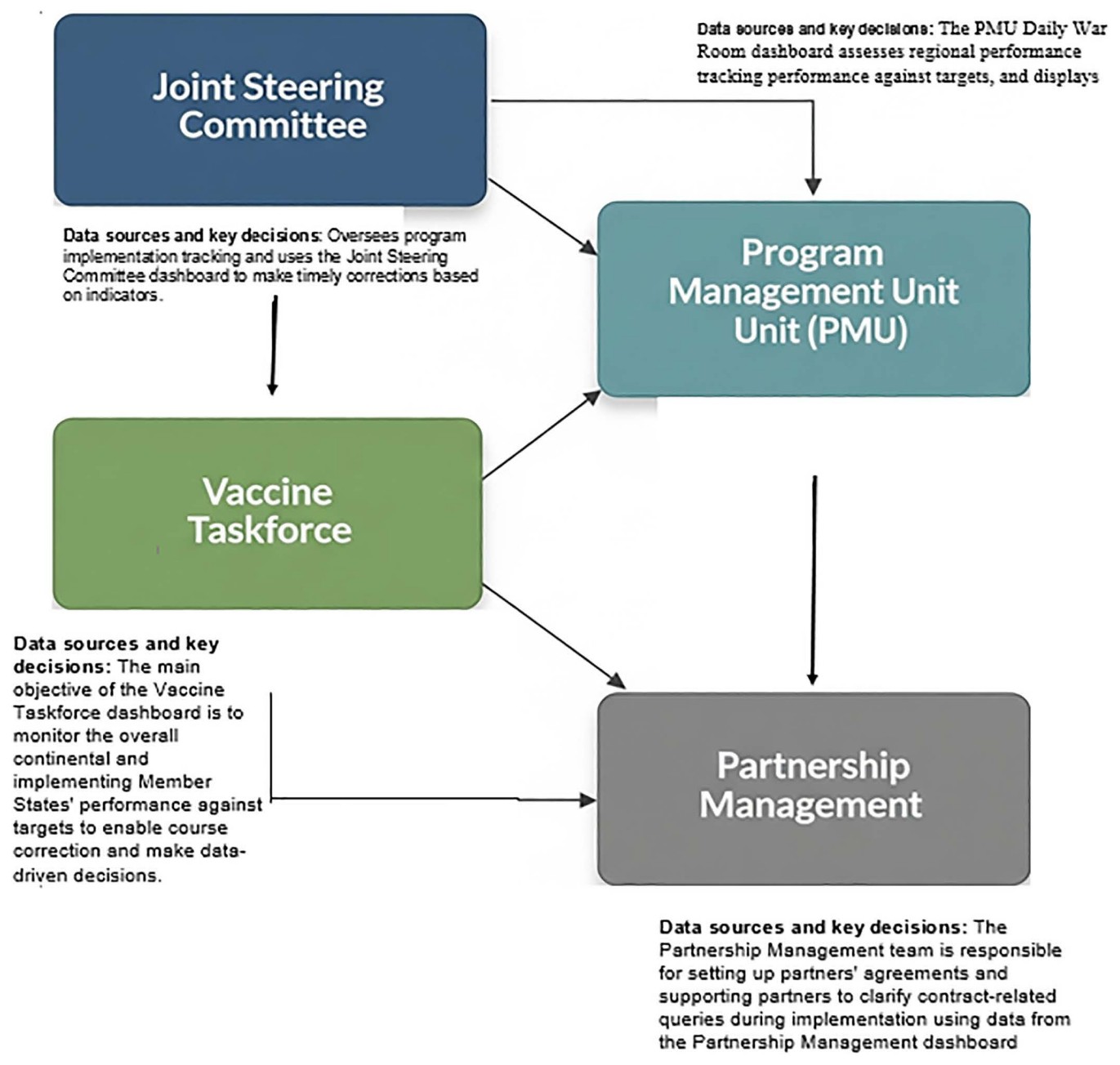

**Fig 8. Schematic diagram for these four bodies, key decisions each makes, and the data sources needed to execute their work.**

## Discussion

The rapid onset of the COVID-19 pandemic necessitated that countries worldwide quickly implement data-reporting systems, many of which relied on digital tools. This urgency, alongside the strain on health systems, led to increased collaboration between innovators and governments to adapt digital technologies for various functions, including case management, contact tracing, evidence-based surveillance, risk communication, and vaccine distribution [19]. This study presents the COVID-19 vaccination data management and visualization systems that enhanced decision-making at the

continental level in Africa. Key insights from the Africa CDC's Saving Lives and Livelihoods (SLL) Program offer valuable lessons for the global community, demonstrating the power of digital solutions in public health management.

## The SLL program's decision-making framework

A significant innovation of the SLL program was establishing four primary decision-making bodies supported by specialized dashboards to facilitate data-driven action. This approach aligns with the broader trend in public health of using data visualization tools to enhance decision-making processes. Research has shown that real-time dashboards are essential for managing health interventions during crises like the COVID-19 pandemic [20] and are crucial for transparency and stakeholder engagement [9]. Thus, the SLL program reflects a global shift towards data-centric governance, leveraging technology to drive effective program outcomes.

At the apex of the SLL program's governance is the Joint Steering Committee, led by Africa CDC and the Mastercard Foundation, focusing on monitoring and adjusting program implementation. This structure is pivotal in meeting vaccination targets, a goal often highlighted in public health studies. For instance, research suggests that effective leadership and monitoring bodies can significantly enhance vaccination coverage [21]. The program's dashboard, with regional breakdowns showcasing performance variations among countries, resonates with findings that disparities in vaccination uptake often stem from differing health system capacities and regional policies [22]. This evidence underscores the need for tailored support to improve vaccination rates across all Member States.

The Program Management Unit (PMU) functions as the operational core of the SLL program, ensuring coordination and progress reporting. The PMU's Daily War Room Dashboard, which tracks vaccination targets and doses administered weekly, aligns with best practices in health intervention management. Continuous performance tracking allows timely adjustments, enhancing overall program outcomes [23]. Additionally, the emphasis on utilizing both permanent and mobile clinics for vaccine administration highlights the importance of accessibility in vaccination campaigns, particularly in underserved regions [24], demonstrating a commitment to comprehensive coverage.

The Vaccine Taskforce Dashboard provides a comprehensive view of vaccination efforts across Member States, enabling data-driven adjustments where needed. This adaptability is supported by studies that emphasize the necessity of flexible frameworks in public health [25], which allow for rapid responses to challenges. The "bird's-eye view" approach of the Taskforce Dashboard facilitates the identification of trends and coverage gaps, fostering proactive decision-making to achieve program objectives. This perspective is supported by research indicating that comprehensive data collection enhances intervention effectiveness [26].

The Partnership Management Dashboard plays a critical role in monitoring partner performance across Member States, ensuring compliance and effective collaboration. Successful health interventions often rely on strong partner relationships and clear communication, with studies showing that effective partnership management correlates with improved program outcomes [27]. The dashboard's emphasis on tracking compliance and supporting agreements highlights the role of robust contractual frameworks in fostering partner accountability [28], which is essential for optimizing health service delivery.

The Monitoring and Evaluation (M&E) component of the SLL program, led by Africa CDC, focuses on collaborative data collection and validation, a trend increasingly advocated in public health to improve data reliability. Inclusive M&E strategies yield more comprehensive insights into program effectiveness [29]. The program's weekly tracking of vaccine administration to maintain accountability and transparency echoes best practices in health monitoring, highlighting the importance of timely reporting for stakeholder trust [30]. This collaborative approach enhances the program's adaptability, strengthening its response to evolving vaccination challenges.

## Lessons learned

The SLL program's strategic use of decision-making bodies and dashboards reflects a commitment to effective program management, accountability, and data-driven decision-making. The COVID-19 vaccination dashboard, developed

by Africa CDC with support from HISP South Africa, has become an essential tool for real-time monitoring of vaccine procurement, administration, and data management, informing Africa CDC and Mastercard leadership on vaccination progress across the continent. The dashboard's interactive features allow for rapid filtering and visualization of data, supporting decision-makers in implementing COVID-19 vaccination strategies at all levels. On average, since June 2022, SLL program implementers have administered approximately 582,000 doses weekly to eligible populations across Member States.

Near real-time data capturing and digital governance have accelerated data sharing, analysis, and utilization, enabling implementers and Member States to monitor vaccination progress, identify gaps, and manage vaccine supply and logistics. This capacity for real-time adjustments has proven vital for vaccine equity and accessibility, essential components of a successful continental vaccination strategy. The COVID-19 vaccination dashboard continues to evolve based on the SLL vaccination protocols, supporting targeted vaccination campaigns and fostering equitable vaccine uptake across Member States. These insights suggest that adaptable, interactive dashboards have the potential to transform the response to public health challenges, enabling effective distribution and uptake of vaccinations and enhancing readiness for future disease outbreaks.

### Challenges

While the Africa CDC COVID-19 vaccination dashboard proved highly valuable, several implementation barriers were encountered. Political and logistical hurdles occasionally slowed adoption, as some Member States prioritized existing national or donor-supported systems, necessitating sustained advocacy and coordination to harmonize approaches. Scalability concerns centered around ensuring consistent dashboard use across multiple countries, which required ongoing training, technical support, and integration with diverse health information systems.

At the operational level, limited internet connectivity, frequent power outages, and dependence on paper-based registers, especially in rural and remote facilities, resulted in data delays, incomplete reporting, and inconsistencies requiring additional validation. Variability in users' data visualization literacy further underscored the need for continuous capacity building. Differences in data formats and reporting processes across countries challenged interoperability and seamless data integration. Moreover, shortages of dedicated data managers placed additional pressure on national and regional coordinators to troubleshoot and maintain data integrity.

Despite these challenges, regular War Room meetings and remote support facilitated prompt identification and resolution of data anomalies, enabling steady improvements in data quality and reporting consistency. These experiences highlight the critical importance of sustained investments in digital infrastructure, workforce development, and governance frameworks to ensure the sustainability and broader scalability of similar digital health systems across Africa.

### Limitations

This manuscript primarily provides a descriptive account of the design and implementation of the Africa CDC DHIS2 COVID-19 vaccination dashboard. It lacks rigorous impact evaluation and causal analysis of vaccination outcomes. The user feedback included is based on internal reports rather than derived from structured qualitative or quantitative studies. Variability in data completeness and quality across Member States may affect uniformity in interpretation. In addition, no cost-effectiveness and interoperability comparison with other digital platforms is presented, which could enhance contextual understanding.

### Conclusion

The SLL program demonstrates the importance of strategic decision-making frameworks and digital dashboards in supporting program management and fostering a data-driven, adaptable approach to public health. By examining these strategies in light of current public health literature, the program emerges as an exemplary model within the paradigm of adaptability, collaboration, and transparency in achieving health objectives. Rapid, multi-level communication between

managers, staff, and implementing partners has enabled the Africa CDC to improve COVID-19 vaccine uptake, manage ongoing challenges, and prioritize Member States' needs. Translating this framework of digital immunization dashboards to other public health areas while leveraging existing digital infrastructure can enable more efficient use of resources, promote interoperability and standardization across countries, particularly in low -and -middle income countries (LMICs), and ultimately enhance health equity and advance digital health initiatives across the continent. Moreover, Integrating digital health solutions with immunization strategies can enhance coverage and monitoring, improving public health outcomes. In the post-COVID-19 era, leveraging digital innovations can make immunization programs more efficient and effective, promoting better health for all.

## Supporting information

**S1 Table. Share of target population fully vaccinated.**
(XLSX)

**S2 Table. SLL - Eastern Region Cumulative performance against target per Member state in %.xls**
(XLSX)

## Acknowledgments

We acknowledge Africa CDC's leadership and expertise in various workstreams, for developing and making use of the COVID-19 vaccination tracking dashboard to ease the decision-making at various structures of the continental health agency. Moreover, we thank the Implementing Partners and Member States for their dedication to submitting COVID-19 data with full quality and integrity.

## Author contributions

**Conceptualization:** Raji Tajudeen, Mosoka Papa Fallah, John Ojo, Tamrat Shaweno, Ngashi Ngongo.

**Data curation:** John Ojo, Wondwossen Amanuel, Michael Sileshi Mekbib, Frehiwot Mulugeta, Moses Bamatura, Dennis Kibiye.

**Formal analysis:** Tamrat Shaweno, Nebiyu Dereje.

**Funding acquisition:** Raji Tajudeen, Mosoka Papa Fallah, Ngashi Ngongo.

**Investigation:** John Ojo, Wondwossen Amanuel, Michael Sileshi Mekbib, Frehiwot Mulugeta, Moses Bamatura, Dennis Kibiye, Patrick Chanda Kabwe, Senga Sembuche, Nebiyu Dereje.

**Methodology:** John Ojo, Tamrat Shaweno, Wondwossen Amanuel, Frehiwot Mulugeta, Moses Bamatura, Dennis Kibiye, Nebiyu Dereje.

**Project administration:** John Ojo.

**Software:** John Ojo, Tamrat Shaweno, Wondwossen Amanuel, Michael Sileshi Mekbib, Frehiwot Mulugeta, Nebiyu Dereje.

**Supervision:** Nebiyu Dereje, Jean Kaseya.

**Validation:** Mosoka Papa Fallah, John Ojo, Frehiwot Mulugeta, Patrick Chanda Kabwe, Senga Sembuche.

**Visualization:** John Ojo, Tamrat Shaweno, Wondwossen Amanuel, Michael Sileshi Mekbib, Moses Bamatura, Dennis Kibiye, Nebiyu Dereje.

**Writing – original draft:** John Ojo, Tamrat Shaweno, Wondwossen Amanuel, Frehiwot Mulugeta, Moses Bamatura, Dennis Kibiye, Patrick Chanda Kabwe.

**Writing – review & editing:** Mosoka Papa Fallah, John Ojo, Tamrat Shaweno, Michael Sileshi Mekbib, Nebiyu Dereje, Jean Kaseya.

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
