## [Decision Letter · Decision Letter 0]

29 Jun 2025

Response to Reviewers
Revised Manuscript with Track Changes
Manuscript
**Journal Requirements:**

1. We note that your Data Availability Statement is currently as follows: All data are in the manuscript files.

2. Please provide separate figure files in .tif or .eps format.

3. Please upload a copy of Figure 1, 2, 3, 4, 5 which you refer to in your text on page 9, 10, 12, 14, 15. Or, if the figure is no longer to be included as part of the submission please remove all reference to it within the text.

4. Some material included in your submission may be copyrighted. According to PLOS’s copyright policy, authors who use figures or other material (e.g., graphics, clipart, maps) from another author or copyright holder must demonstrate or obtain permission to publish this material under the Creative Commons Attribution 4.0 International (CC BY 4.0) License used by PLOS journals. Please closely review the details of PLOS’s copyright requirements here: PLOS Licenses and Copyright. If you need to request permissions from a copyright holder, you may use PLOS's Copyright Content Permission form.

Potential Copyright Issues:

Figures 2 and 4: please (a) provide a direct link to the base layer of the map (i.e., the country or region border shape) and ensure this is also included in the figure legend; and (b) provide a link to the terms of use / license information for the base layer image or shapefile. We cannot publish proprietary or copyrighted maps (e.g. Google Maps, Mapquest) and the terms of use for your map base layer must be compatible with our CC-BY 4.0 license. 

**Additional Editor Comments:**
**Reviewers' Comments:**

**Comments to the Author**

1. Does this manuscript meet PLOS Digital Health’s publication criteria?

Reviewer #1: Yes

Reviewer #2: Yes

2. Has the statistical analysis been performed appropriately and rigorously?

Reviewer #1: N/A

Reviewer #2: N/A

3. Have the authors made all data underlying the findings in their manuscript fully available (please refer to the Data Availability Statement at the start of the manuscript PDF file)?

Reviewer #1: Yes

Reviewer #2: No

4. Is the manuscript presented in an intelligible fashion and written in standard English?

Reviewer #1: Yes

Reviewer #2: Yes

Reviewer #1: The paper details the development of a continental COVID-19 vaccination tracking system by Africa CDC and HISP-SA using DHIS2. The system enables real-time monitoring of vaccine distribution, with 300 users accessing role-based dashboards. It provides high-level summaries for leadership and detailed data for programmatic users, supporting informed decision-making. As COVID-19 remains a concern, the system continues to track vaccine uptake and related metrics. Its success offers a model for managing public health data on a large scale.

Need to check on data availability as there was no attachment or link seen.

Some sentences are overly complex and could be revised for clarity. There are minor grammatical issues and typos that should be corrected.The use of technical terms should be consistent and well-defined to ensure clarity for a broad readership.

The research topic is highly relevant and timely and the manuscript is generally well-structured.

Reviewer #2: This paper is timely and a good summary of the seminal efforts that went to into digitizing the support required to power vaccination delivery and results monitoring. Having said that, the manuscript would be better if certain additional details were provided. This will be esp. beneficial for the readers and other stakeholders in LMIC settings who would like to replicate such efforts in their public health delivery programs. Listing them below:

1. Please provide some specific context of the challenges encountered by the stakeholders before the data management system and dashboard were in place. E.g. it took the joint steering committee 1 month to receive upto date information to make key decisions, before digitization and near real time updates.

2. How long did it take for the data management system to go live and become functional? from concept to production?

3. What were the reasons for choosing the DHIS2 platform? E.g. its open source, handles multiple data sources, works in offline settings, can be customized and allows building apps on it.

4. what were the highest level KPIs in tracking the buildout and rollout of this data management system? was it done in phases? if so, how was it done?

5. one of the key takeaways on the paper is the governance structure about the 4 decision making bodies- joint steering committee, program management unit, vaccine taskforce and partnership management. Please provide a schematic diagram for these four bodies, key decisions each makes and the data sources + information needed to execute their work. It will be easier for the readers to grasp it visually.

6. What if any, would be best practice recommendations and practical tips for agencies in other LMIC settings to build similar digital infrastructure for their public health initiatives.

**Do you want your identity to be public for this peer review?** For information about this choice, including consent withdrawal, please see our Privacy Policy

Reviewer #1: No

Reviewer #2: **Yes: ** Rutwik Shah

**Figure resubmission:****Reproducibility:** To enhance the reproducibility of your results, we recommend that authors of applicable studies deposit laboratory protocols in protocols.io, where a protocol can be assigned its own identifier (DOI) such that it can be cited independently in the future. Additionally, PLOS ONE offers an option to publish peer-reviewed clinical study protocols. Read more information on sharing protocols at https://plos.org/protocols?utm_medium=editorial-email&utm_source=authorletters&utm_campaign=protocols

---

## [Decision Letter · Decision Letter 1]

29 Aug 2025

Response to Reviewers
Revised Manuscript with Track Changes
Manuscript
**Journal Requirements:**
**Additional Editor Comments (if provided):**

Thank you for your resubmission and addressing the initial set of comments. One reviewer notes that they would be interested in any type of performance metric, if there are any available. We recognize that this may not have been part of the implementation process, but if there is evidence of impact, that would strengthen the manuscript. Further, this reviewer notes that some discussion of limitations and challenges would also be helpful. Finally, they noted some typographical issues.

Thank you

Reviewer #1:

Reviewer #2:

**Reviewers' Comments:**

**Comments to the Author**

Reviewer #1: All comments have been addressed

Reviewer #2: All comments have been addressed

Reviewer #3: (No Response)

publication criteria?

Reviewer #1: Yes

Reviewer #2: Yes

Reviewer #3: Partly

3. Has the statistical analysis been performed appropriately and rigorously?

Reviewer #1: N/A

Reviewer #2: N/A

Reviewer #3: N/A

4. Have the authors made all data underlying the findings in their manuscript fully available (please refer to the Data Availability Statement at the start of the manuscript PDF file)?

Reviewer #1: Yes

Reviewer #2: Yes

Reviewer #3: No

5. Is the manuscript presented in an intelligible fashion and written in standard English?

Reviewer #1: (No Response)

Reviewer #2: Yes

Reviewer #3: Yes

Reviewer #1: Comments addressed

Reviewer #2: Updates to the manuscript are satisfactory

Reviewer #3: See the attachment.

**Do you want your identity to be public for this peer review?** For information about this choice, including consent withdrawal, please see our Privacy Policy

Reviewer #1: No

Reviewer #2: No

Reviewer #3: No

**Figure resubmission:**

**Reproducibility:** To enhance the reproducibility of your results, we recommend that authors of applicable studies deposit laboratory protocols in protocols.io, where a protocol can be assigned its own identifier (DOI) such that it can be cited independently in the future. Additionally, PLOS ONE offers an option to publish peer-reviewed clinical study protocols. Read more information on sharing protocols at https://plos.org/protocols?utm_medium=editorial-email&utm_source=authorletters&utm_campaign=protocols

---

## [Decision Letter · Decision Letter 2]

20 Nov 2025

COVID-19 Vaccination Data Management and Visualization Systems for Improved Decision-Making: Lessons Learnt from Africa CDC Saving Lives and Livelihoods Program

PDIG-D-25-00092R2

Dear Mr Shaweno,

We are pleased to inform you that your manuscript 'COVID-19 Vaccination Data Management and Visualization Systems for Improved Decision-Making: Lessons Learnt from Africa CDC Saving Lives and Livelihoods Program' has been provisionally accepted for publication in PLOS Digital Health.

Best regards,

Hanieh Razzaghi

Section Editor

PLOS Digital Health

**Additional Editor Comments (if provided):**

Thank you for your excellent manuscript. My suggestion to the authors is to edit the abstract to include some of the quantitative output that was integrated with the Results section in the last round of edits, which I think will significantly strengthen the manuscript.

**Reviewer Comments (if any, and for reference):**

Reviewer's Responses to Questions

**Comments to the Author**

Reviewer #3: All comments have been addressed

publication criteria?

Reviewer #3: Partly

3. Has the statistical analysis been performed appropriately and rigorously?

Reviewer #3: No

4. Have the authors made all data underlying the findings in their manuscript fully available (please refer to the Data Availability Statement at the start of the manuscript PDF file)?

Reviewer #3: Yes

5. Is the manuscript presented in an intelligible fashion and written in standard English?

Reviewer #3: Yes

Reviewer #3: This revised version shows clear improvement in organization and clarity. The paper provides a valuable and timely account of how Africa CDC leveraged a DHIS2-based data management and visualization platform to support COVID-19 vaccination tracking and decision-making across the continent. The system description is coherent and well contextualized within Africa’s public health infrastructure.

To further strengthen the paper, I suggest a light language edit to improve flow and readability, especially in the “Lessons Learned” section.

**Do you want your identity to be public for this peer review?** For information about this choice, including consent withdrawal, please see our Privacy Policy

Reviewer #3: No
